

# Application of open field, tonic immobility, and attention bias tests to hens with different ranging patterns

Dana L.M. Campbell[1], Emily J. Dickson[1,2] and Caroline Lee[1]

[1] Agriculture and Food, Commonwealth Scientific and Industrial Research Organisation (CSIRO), Armidale, New South Wales, Australia

[2] School of Environmental and Rural Science, University of New England, Armidale, New South Wales, Australia

## ABSTRACT

Assessment of negative affective states is a key component of animal welfare research. In laying hens, excessive fearfulness results in reduced production and increased sensitivity to stress. Fearfulness can be defined as a response to a known threat, but anxiety is a response to an unknown threat and may have similar negative consequences. The open field test and tonic immobility test are commonly applied to measure fearfulness in laying hens. An attention bias test that measured individual hen's responses to playback of a conspecific alarm call in the presence of food was recently pharmacologically validated using an anxiogenic drug but was confounded by the hen's typical motionless response in a novel environment. The current study used 56-week old free-range layers to further assess the validity of an attention bias test to differentiate ranging treatment groups in comparison with the open field and tonic immobility tests. The selected hens varied in their range use patterns as tracked by radio-frequency identification technology. 'Indoor' hens did not access the range and 'outdoor' hens ranged daily; previous research has confirmed higher fearfulness in hens that remain indoors. The tonic immobility test did not differentiate ranging groups ($P = 0.34$), but indoor birds were slower to first step ($P = 0.03$) and stepped less ($P = 0.02$) in the open field test. The attention bias test occurred in an isolated wooden box using a conspecific alarm call playback (a threat) and mixed grain (a positive stimulus). The behavioural response of latency to resume eating following playback of the alarm call was measured to differentiate the anxiety states of the indoor and outdoor ranging birds. Before the attention bias test could occur, birds had to be habituated to the test box across three separate 5-minute sessions to increase the willingness to feed within the novel test environment. All birds ate faster across time ($P < 0.001$) but the indoor birds were slower to eat than the outdoor birds ($P < 0.001$). In this study, the latency to resume eating following an alarm call was determined to be a poor measure for highly anxious birds as they failed to eat at all. Forty-six percent of indoor hens were excluded for not eating across the 5-minute test. Of the birds that did eat, only 7% of indoor hens ate following playback of the alarm call, compared with 36% of outdoor hens. This repetition of an attention bias test for laying hens highlights the challenges in assessing hens with extreme fearful/anxious responses and that information may be missed when non-performing hens are excluded from behavioural tests. We suggest that latency to eat in a novel arena without any alarm call playback is an informative measure of

Corresponding author
Dana L.M. Campbell,
Dana.Campbell@csiro.au

anxious state that can be applied to all hens but consideration must be made of potential differences in food motivation.

## INTRODUCTION

Affective state is a key component of animal welfare assessment (*Duncan, 2005*). In particular, negative affective states such as fear and anxiety can be an indication of how an animal is coping within its environment (*Duncan, 2005*). In commercial laying hens, excessive fear has been linked with increased sensitivity to stress, decreased production, lower body weight and poorer feed intake (*De Haas et al., 2012*; *De Haas et al., 2013*); placing the hens in a negative welfare state. Behavioural tests such as the open field test and tonic immobility are widely used to classify hen's fearfulness. The locomotory and vocal behaviours exhibited in the open field test are believed to be a conflict between fear of a novel environment which reduces locomotion and desire for social reinstatement which increases vocalisations (*Gallup & Suarez, 1980*; *Vallortigara, 1988*); the duration of tonic immobility is classified as a predator avoidance strategy (*Gallup, Nash & Ellison, 1971*; *Jones, 1986*). However, animals may also experience anxiety which in excess, can have similar negative welfare consequences to fear (*Ohl, Arndt & Van Der Staay, 2008*). It can be challenging to distinguish between fear, defined as a response to a known threat, and anxiety which is a response to a potential or unknown threat (*Steimer, 2002*). For laying hen welfare assessment, some researchers have used the open field test as an assessment of anxiety (*Nordquist et al., 2011*) or desire for social reinstatement following isolated placement into a novel environment (*De Haas et al., 2014*), but anxiety is not as commonly assessed or differentiated from fearfulness (*Forkman et al., 2007*).

Cognitive bias tests provide a method of assessing affective states, as differing emotional states can influence cognitive processes such as judgement and attention (*Mendl et al., 2009*). Judgement bias tests have become a popular method of measuring optimistic and pessimistic cognitive biases but can be very time-consuming, often requiring extensive training of test subjects (*Bethell, 2015*). Additionally, the results can be contextually variable and inconsistent. Chicks in an isolation-induced anxious state showed a pessimistic bias towards visual predator cues (owl silhouette; *Salmeto et al., 2011*) but socially-isolated adult hens showed no judgement bias towards ambiguous food reward cues (*Hernandez et al., 2015*) and individual differences in fearfulness can affect the ability of hens to learn judgement bias tasks (*De Haas, Lee & Rodenburg, 2017*). Unexpected findings have been reported with an optimistic bias found in response to environmentally-induced chronic stress (*Verbeek et al., 2019*) and pharmacologically-induced chronic stress did not influence judgement bias in sheep (*Monk, Belson & Lee, 2019*). Alternatively, attention bias has been gaining interest as a valid measure of anxiety in animals where most studies to date have differentiated treatment groups based on their attention biases (reviewed in

*Crump, Arnott & Bethell, 2018*). Humans and other animals will show increased attention towards a threatening situation when they are in a heightened state of anxiety (*Crump, Arnott & Bethell, 2018*; *Shechner et al., 2012*). Attention paid towards a positive stimulus in the presence of a threat has been pharmacologically validated as a short (under 5 min) behavioural test to measure anxiety in sheep and cattle (*Lee et al., 2016*; *Lee et al., 2018*; *Monk et al., 2018a*). An attention bias test was also recently applied to laying hens with pharmacological validation using the anxiogenic drug *meta*-Chlorophenylpiperazine (*m*-CPP). The test was conducted on young adult hens (23 weeks of age) individually tested in an isolated box arena with food present and playback of a conspecific alarm call as a threat (*Campbell et al., 2019*). Those hens that were dosed with *m*-CPP were slower to feed following playbacks of alarm calls and showed a higher rate of stepping and vocalising in the arena in comparison with saline-dosed hens (*Campbell et al., 2019*). However, these hens had to first be habituated to the test arena because a common response of laying hens to a novel environment is to remain stationary or 'freeze' (*Gallup & Suarez, 1980*). An attention bias test cannot work in this situation because it requires the birds to move and eat to be able to measure their responses specifically to the alarm call threat. If the birds are kept in group housing, then individual responses to the auditory playback cannot be measured in the home cages. In an earlier application of the attention bias test to older adult free-range hens (51 weeks of age), the hens did not require habituation to the testing arena and the test was able to behaviourally distinguish between hens of different range use patterns (ranging regularly versus remaining indoors, (*Campbell et al., 2019*). It was not clear if the differences between the two testing situations were a result of hen age, strain, or previous experiences with the older hens rapidly acclimating to the testing arena. Favati and colleagues (*2014*) successfully applied an alarm call playback to male Swedish bantams in the presence of food ('startle test') to differentiate dominance status, but the birds were observed for 20 min in the novel test arena before the playback. Similarly, Schütz and colleagues (*2001*) presented a hawk model to chickens in a test arena containing food following 10 min of initial adaptation. Further application of an individual-bird attention bias test using an alarm call playback for welfare assessment is required.

Free-range hens have a choice of accessing the range environment or not and are well-known to vary individually in the amount of time they spend outdoors (e.g., *Campbell et al., 2016*; *Hartcher et al., 2016*; *Larsen et al., 2018*). Some hens will range daily but other hens will remain inside. Behavioural tests of open field, tonic immobility, novel object, and human avoidance across several different flocks have shown correlational relationships between range use and fear with hens that remain indoors being more fearful (*Campbell et al., 2016*; *Hartcher et al., 2016*; *Larsen et al., 2018*). These extreme behavioural ranging groups thus represent known treatment groups for further assessing the potential for an attention bias test to differentiate anxiety states in laying hens.

The aims of this study were to apply the attention bias test to adult free-range laying hens that differed in their range use patterns to determine if it could be refined in its application with a shorter duration of total testing time including how the test compared with the commonly used open field and tonic immobility tests. It was predicted that hens that did

not range would show greater fear and anxiety responses across the tests and that the older hens would rapidly acclimate to the test arena.

## MATERIALS & METHODS

### Ethical statement
Research was approved by the University of New England Animal Ethics Committee (AEC17-092).

### Animals and housing
A total of 67 HyLine[®] Brown hens at 56 weeks of age were selected from a total flock of 1386 hens that were housed in an experimental free-range facility. The hens were part of a larger study that investigated the impacts of three different rearing enrichments on the behaviour, welfare, and production of free-range hens across a flock cycle. All pullets were reared at a separate facility from day-old chicks until 16 weeks of age. They were housed in nine separate pens across three rooms and exposed to either (1) novel objects that changed weekly, (2) custom-designed perching structures, or (3) just floor litter (control). The pullets were transferred to the laying facility at 16 weeks of age and treatment groups were housed under the same conditions henceforth. Pullets were kept in pens within their rearing treatments (9 pens, 3 replicates per rearing treatment) but to minimise variation, only hens from the control treatment pens ($n = 3$) were selected for this study.

The free-range layer facility contained 9 floor litter pens within a single shed that were visually isolated via shade cloth. Each pen (4.8 m L × 3.6 m W) held 154 hens at an approximate indoor stocking density of nine birds per m$^2$ with available nest box space, perches, suspended round feeders and water nipple resources to meet the Australian Model Code of Practice for the Welfare of Animals –Domestic (*Primary Industries Standing Committee, 2002*). Commercial mash feeds formulated for appropriate bird ages were provided *ad libitum* throughout the lay cycle. The LED lighting schedule gradually increased to 16 h light and 8 h dark by 30 weeks of age. The shed was fan-ventilated only with no temperature or humidity control.

Each indoor pen connected to a separate outdoor range area with no trees, bushes or shelter and ranges were visually separated by shade cloth along the fence lines. At 25 weeks of age (May 2018), hens were provided first access to the outdoor area. Daily access was granted via automated pop-holes from 0915 h until closing after sunset. Range access hours equated to approximately 8 h across winter and then approximately 11 h when daylight savings time started (October 2018). The ranges were initially grassed but were mostly bare dirt at the time of testing in the current study.

### Radio-frequency identification, plumage coverage, and hen selection
Radio-frequency identification (RFID) systems were set up within each pop hole with two passageways per pen to allow movement between the range and indoor pen. The RFID systems were designed and supported by Microchips Australia Pty Ltd (Keysborough, VIC, Australia) with equipment developed and manufactured by Dorset Identification B.V. (Aalten, the Netherlands) using Trovan[®] technology. A schematic of the RFID system

is available in *Campbell et al. (2018)*. Each hen was banded with a microchip (Trovan®Unique ID 100 (FDX-A): operating frequency 128 kHz) glued into an adjustable leg band (Roxan Developments Ltd, Selkirk, Scotland). As banded hens passed over the antennas the system recorded the time and date of each bird passing through and in which direction (onto the range, or into the pen) at a precision of 0.024 s (maximum detection velocity 9.3 m/s). Individual ranging data were collected daily from 25 weeks of age onwards. RFID data from 49 until 53 weeks of age (25 days of data) for all hens from the control pens were initially run through a custom-designed software program written in the 'Delphi' language (Bryce Little, CSIRO, Agriculture and Food, St Lucia, QLD, Australia) that filtered out any unpaired or 'false' readings. These can occur if a hen sits in a pop-hole or does not complete a full transition between the indoor pen and range area. Once filtered, the program summarised the daily time spent outdoors for each hen across the selected period. From the three pens, 67 birds (25, 21, and 21 from the three pens respectively) were selected based on extreme ends of the individual ranging variation. Thirty-one hens who rarely used the range were identified (indoor hens, the limiting factor in sample size selection), and 36 hens who used the range daily for relatively higher amounts of time within the pen were selected. Hens from both ranging groups were selected from each of the three pens. Extra outdoor hens were included in case of injury, mortality, or difficulty in finding them again within the pen. However, none of these issues occurred and all 36 hens were able to be tested.

Hens were also identified based on plumage coverage as scored by a single experimenter blinded to range use patterns. The original sample intention was to select hens across both ranging groups that also varied in their plumage coverage as scored using the LayWel scoring system (*LayWel, 2006*) at 54 and 58 weeks of age. This system scores the neck, back, chest, wings, tail, and vent from 1 to 4, with a score of 4 indicating good plumage coverage (maximum score of 24 across all body parts). In this study, only the back of the neck was scored as feather damage on the front of the neck was observed to be more likely due to rubbing on the feeder rims rather than feather pecking. However, the majority of birds displaying feather pecking damage were also indoor hens. Thus, the selected birds were outdoor hens with good plumage coverage (mean score 23.49 ± SE 0.14 of a maximum 24) and of all the indoor hens, we selected those with poorer plumage coverage (mean score 21 ± SE 0.21 of a maximum 24). There were some indoor hens that had good feather coverage and some outdoor hens with poorer plumage but of insufficient numbers to include as treatment groups and they were not selected for testing. We focussed on feather-pecked indoor ranging hens versus outdoor ranging hens of good plumage coverage to ensure we had the most extreme sample groups for further validation of the attention bias test.

Prior to behavioural testing, the selected birds were all banded with extra leg bands for easy detection within their home pens.

## Behavioural testing

All testing occurred in a separate enclosed room adjacent to the main housing shed. Testing occurred from approximately 10:00 until 17:30 on each testing day. All birds were still

permitted range access during the testing days so that normal ranging behaviour could proceed outside of the specific period of testing. Most birds were able to be captured while inside the shed with a few outdoor birds captured from the range area as necessary. The arena used for the open field test, habituation sessions, and attention bias test was a wooden square box (1.8 m × 1.8 m), with a shade cloth ceiling including a gap at the top to allow video-recording. The shade-cloth on the top of the testing arena reduced the brightness of the testing environment so that it was visually similar to the indoor pens when the pop-holes were open (specific lux measurements were not taken). Two video cameras (Sony Handycams HDR-PJ410, Sony Corporation, Tokyo, Japan) were mounted directly above the testing arena, one which recorded bird behaviour and the other which connected to a screen for real-time observation. There was a small hen-sized opening flap on one side of the arena for placing the hen into the testing area, and one side of the arena opened up for removal of the hen after testing.

At 56 weeks of age, all birds were tested in an open field test within the arena across two days. Individual hens were selected from home pens as they were first identified, carried to the testing room and placed into the arena with the lights off. The lights were switched on and the test immediately began with a single observer present in the room but out of sight. The latency to first move (more than just their head but did not take a step), step, vocalise, and number of individual vocalisations per test minute were all recorded live by the same trained experimenter, blind to the hen's ranging classification. The test concluded after 5 min, the bird was fitted with an additional leg band to mark testing completion and immediately returned to its home pen. Hens were selected in rotating order across the three pens until all testing was complete. The video was later observed by a single trained and blinded experimenter to count the number of steps per test minute that each hen made in the arena.

Following the open field test, the hens were given additional habituation sessions to the test arena before the attention bias test was conducted. This was to minimise the hens responding to the arena itself versus the intended threat (conspecific alarm call playback). Mixed grain was added in a small pile at the centre wall opposite to the entry flap for the habituation sessions. Some grain had been placed in pen feeders twice prior to the habituation sessions to acclimate the hens to the novel food. It was logistically difficult to record if all test hens had consumed the grain which may have resulted in different levels of acclimation to the novel feed. At 56 and 57 weeks of age, each hen was caught and placed into the arena in the same process as the open field test and given 5 min to acclimate. An experimenter recorded the latency to move, step, vocalise, and eat. This was repeated three times on separate days; each hen had been in the arena for a total of 20 min prior to attention bias testing.

Prior to the attention bias testing it was intended that all hens were consuming food in less than one minute following placement in the arena (similar to *Campbell et al., 2019*) and to be able to measure the food-related response to the alarm call). At the end of habituation, the majority of outdoor hens (outdoor: 24/36, indoor: 11/31) had achieved this, but in contrast, 16/31 indoor hens were still not eating following their habituation sessions (outdoor: 5/36). The decision was made to progress with the attention bias test

with all hens as further habituation sessions may have had minimal impact on the indoor hens' willingness to eat.

The attention bias test was conducted following the same protocol as the habituation sessions with the addition of a pre-recorded short conspecific alarm call played 5 s after the bird first ate. Alarm calls were as per those used in *Campbell et al. (2019)*. They had been opportunistically recorded using a Roland R-05 MP3 recording device (Roland Corporation US, Los Angeles, CA, USA) with a sampling rate of 48 kHz, and a bit rate of 128 kbps from a different group of 100 ISA Brown hens that were disturbed by a wild bird outside their shed. The audio clips were edited into short playback files using the software Audacity 2.0.3 (http://audacityteam.org/download/). The alarm calls were played from a JBL Clip 2 portable Bluetooth speaker at approximately 75–80 dB (JBL, Los Angeles, CA, USA). Non-tested hens could not hear the alarm calls as testing occurred in a separate room and the calls were not able to be heard through the testing box and room walls. Experimenters could not hear the call playbacks in the main shed and no visible reaction from hens in their home pens were observed when the alarm calls were played during testing. There were anecdotal observations of multiple occurrences across the flock cycle of something unknown disturbing the hens inside and all birds erupting into alarm calling. This often occurred while all hens were inside and thus all birds would have been exposed (but in potentially varying degrees) to conspecific alarm calls of their flock prior to testing. The latency to step, vocalise, and eat both before and after the alarm call were recorded. Two separate alarm calls were alternated between tested birds. The test concluded after 5 min. Five hens were removed from testing as they had been used to trial a different threatening visual stimulus before deciding to remain with the audio playback.

Finally, a tonic immobility test was also conducted on all hens at 58 weeks of age. Hens were captured from their home pen and carried to the adjacent room. Hens were placed on their backs in a metal cradle and restrained for 5 s by the experimenter placing one hand on the bird's chest and another over its head with the head hanging down. The single experimenter then removed their hands and stepped aside with eyes averted downwards. The test concluded after 5 min of immobilisation or when the bird righted itself after at least 10 s of immobilisation, whichever occurred first. Duration of tonic immobility was recorded, and the restraining was repeated up to 5 times if the hen righted itself in less than 10 s. All hens were feather-scored again by the same experimenter following conclusion of the testing.

## Data and statistical analyses

All analyses were conducted in JMP® 14.0 (SAS Institute, Cary, NC, USA) with $\alpha$ set at 0.05. For the open field test, the latency (seconds) to first move, step, and vocalise were compiled per individual bird ($n = 67$) and $\log_{10}$ transformed. Birds that did not perform the specified behaviour were assigned a value equal to the maximum test time. The number of steps and number of vocalisations per bird for each minute of the test were square-root transformed. The averaged feather score data per bird were square-root transformed. All variables were first tested in a General Linear Model (GLM) for a relationship with feather score separately for indoor and outdoor hens due to the differences in feather score
variation between the two ranging groups. Feather score had a trend for an effect on the latency to step ($P = 0.07$) and the latency to vocalise ($P = 0.06$) within the indoor group but no significant effect on the latency to move and number of steps and vocalisations (all $P \geq 0.10$) and no effect on any variables within the outdoor group (all $P \geq 0.59$). As range use was the factor of primary interest and the impact of feather scores could not be tested in a balanced manner across ranging groups, the effect of feather score was not included in any subsequent models. General Linear Mixed Models (GLMM) were applied to test the fixed effect of ranging group on the latencies, or fixed effects of ranging group and time including their interaction on the step and vocalisation counts with bird ID nested within pen as a random effect. Restricted maximum likelihood estimation methods were applied. Non-significant interactions were removed from the final models. Where significant differences were present, post-hoc Student's t-tests were applied to the least squares means.

The duration (seconds) of tonic immobility was compiled per individual bird ($n = 67$) and $\log_{10}$ transformed. The number of attempts per bird to induce tonic immobility were square-root transformed. GLMs were applied to show no significant relationship between duration and feather score separately within each ranging group (both $P \geq 0.32$). A GLMM was then applied to test the fixed effect of ranging group with bird ID nested within pen as a random effect on the duration of tonic immobility and number of attempts to induce it.

Ranging group was the primary focus for the analyses of the habituation sessions and attention bias test, thus any effect of feather score was not tested. The latencies (seconds) to first move, step, vocalise, and eat were compiled per bird ($n = 67$) for the three habituation sessions and $\log_{10}$ transformed. GLMMs were applied to test the fixed effects of ranging group and habituation session, including their interaction, with bird ID nested within pen as a random effect. Non-significant interactions were removed from the final models and post-hoc Student's t-tests applied to significant effects.

Data from the attention bias test was compiled per bird ($n = 62$) for the latency (seconds) to first step, vocalise, and eat. Data were $\log_{10}$ transformed and the fixed effect of range use was tested using a GLMM with bird ID nested within pen as a random effect. Not every bird received an alarm call playback as not every bird ate. Therefore, the latency to step, vocalise, and eat following the alarm call was assessed for $n = 43$ hens ($n = 15$ indoor, $n = 28$ outdoor) only (i.e., 43/62 hens ate food and received an alarm call playback). The maximum remaining test time following the end of the alarm call was assigned for those birds that did not perform these behaviours. The latency data were $\log_{10}$ transformed and analysed using a GLMM with the fixed effect of ranging group and bird ID nested within pen as a random effect.

The raw values are presented in the tables and figures as there was no difference between the raw and back-transformed means.

## RESULTS

The indoor hens took significantly longer to first step ($F_{(1,65)} = 4.74$, $P = 0.03$) but not to first move ($F_{(1,65)} = 2.81$, $P = 0.10$), or first vocalise ($F_{(1,65)} = 0.70$, $P = 0.41$, Table 1). The

**Table 1   Time to perform and quantity of behaviours in an open field test for indoor and outdoor free-range hens.** The mean ± SEM latencies (seconds) to first move, step, and vocalise and number of steps and vocalisations in an open field test for hens that ranged daily (outdoor) or did not range (indoor).

|  | Lat. move (s) | Lat. step (s) | Lat. vocalise (s) | # steps | # vocals |
|---|---|---|---|---|---|
| Indoor hens | 46.77 ± 13.02 | 253.84 ± 16.14[a] | 228.65 ± 22.11 | 1.25 ± 0.35[b] | 1.05 ± 0.25 |
| Outdoor hens | 23.92 ± 4.47 | 181.92 ± 19.58[b] | 192.47 ± 21.22 | 3.1 ± 0.43[a] | 2.22 ± 0.73 |

Notes.
a, b: Dissimilar superscript letters indicate significant differences at $P < 0.05$. Analyses were conducted on transformed data but raw values are presented.

indoor hens made fewer steps than the outdoor hens ($F_{(1,65)} = 5.52$, $P = 0.02$, Table 1) with the number of steps increasing across time for all hens ($F_{(4,264.1)} = 5.10$, $P = 0.0006$) but there was no interaction between ranging group and time ($P = 0.64$). There was no effect of ranging group ($F_{(1,65)} = 0.64$, $P = 0.43$), or time ($F_{(4,264.3)} = 0.55$, $P = 0.70$) on the number of vocalisations (Table 1) and no significant interaction ($P = 0.83$). There were no differences between ranging groups in the duration of tonic immobility ($F_{(1,64)} = 2.08$, $P = 0.15$; mean ± SEM indoor: 150.42 ± 19.02s, outdoor: 115.44 ± 17.63s), or the number of attempts to induce it ($F_{(1,65)} = 1.68$, $P = 0.20$; mean ± SEM indoor: 2.13 ± 0.23, outdoor: 2.53 ± 0.22).

The indoor hens were slower to first move ($F_{(1,65)} = 9.80$, $P = 0.003$), first step ($F_{(1,65)} = 21.85$, $P < 0.0001$), and first vocalise ($F_{(1,65)} = 7.77$, $P = 0.007$) across the habituation sessions and all birds were faster to perform each behaviour with increasing habituation sessions ($F_{(2,132)} = 40.74$–$55.68$, $P < 0.0001$, Fig. 1). There were no interactions between ranging group and habituation sessions (all $P \geq 0.56$). There was however, an interaction between ranging group and habituation session for the latency to eat ($F_{(2,130)} = 3.83$, $P = 0.02$) with the outdoor hens eating faster across time than the indoor hens (Fig. 2). Overall, the indoor hens were slower to eat ($F_{(1,65)} = 12.77$, $P = 0.0007$, Fig. 2). At the end of the habituation sessions 67% (24/36) of outdoor hens were eating but only 48% (15/31) of indoor hens were eating.

For the attention bias test (the fifth time in the test arena), the indoor hens were still slower to first step ($F_{(1,60)} = 24.56$, $P < 0.0001$ mean ± SEM indoor: 129.89 ± 24.95 s, outdoor: 23.32 ± 10.20 s), vocalise ($F_{(1,60)} = 20.09$, $P < 0.0001$ mean ± SEM indoor: 52.61 ± 18.27 s, outdoor: 5.38 ± 1.04 s), and eat ($F_{(1,60)} = 8.43$, $P = 0.005$, mean ± SEM indoor: 169.71 ± 24.52 s, outdoor: 85.56 ± 19.04 s). Following playback of the alarm call there were no differences between range use groups in the latency to step ($F_{(1,41)} = 1.04$, $P = 0.31$, mean ± SEM indoor: 204.93 ± 19.93 s, outdoor: 160 ± 19.57 s), vocalise ($F_{(1,41)} = 0.06$, $P = 0.80$, mean ± SEM indoor: 154.4 ± 24.33 s, outdoor: 151.61 ± 20.32 s) or return to eating ($F_{(1,41)} = 0.65$, $P = 0.43$, mean ± SEM indoor: 209.73 ± 19.92 s, outdoor: 181.25 ± 18.63 s). However, not all birds received an alarm call playback as a total of 19 birds ($n = 13/28$ indoor, $n = 6/34$ outdoor) did not eat. Thus, 46% of indoor hens and 18% of outdoor hens did not receive an alarm call playback (Table 2). Of those that did receive an alarm call, 93% of indoor hens and 61% of outdoor hens did not eat after the alarm. Only 1 indoor hen (of $n = 15$) ate after the alarm call playback, but 11 outdoor hens (of $n = 28$) did (Table 2).

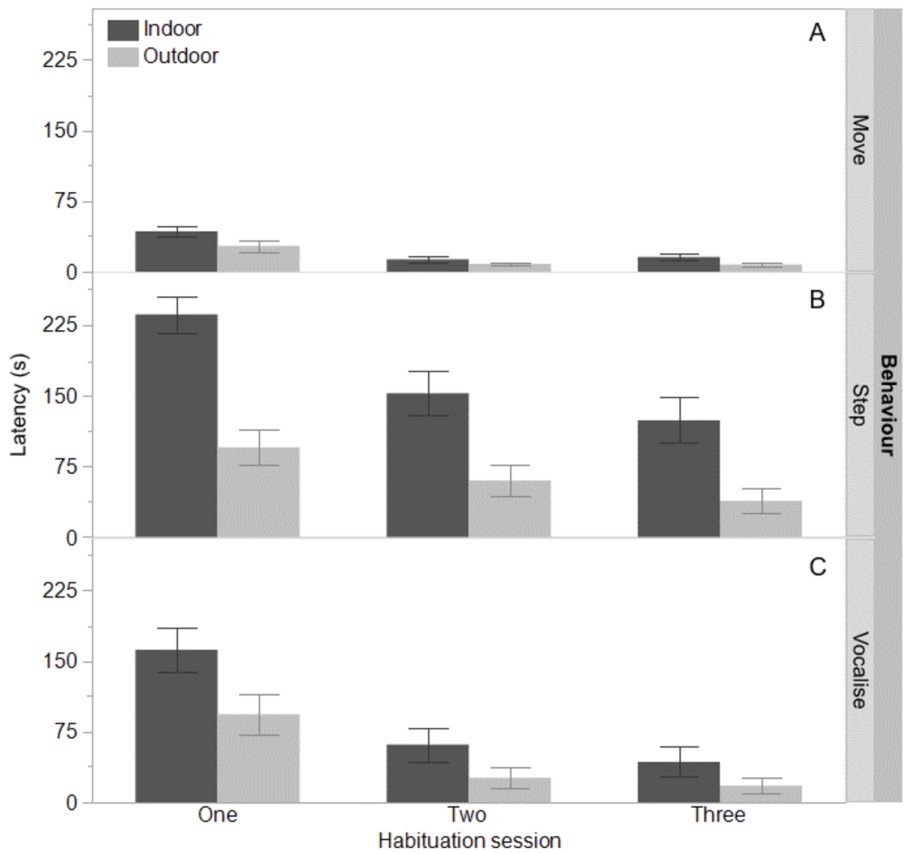

**Figure 1** **Time to move, step, and vocalise for indoor and outdoor free-range hens.** The mean latencies (seconds) ±SEM to first move (A) step (B) and vocalise (C) across three habituation sessions (300 s each) in a test arena for hens that ranged daily (outdoor) or did not range (indoor).

# DISCUSSION

The aims of this study were to refine an attention bias test for chickens by applying it to adult free-range laying hens that either frequently visited the range or remained indoors, while comparing the results with the commonly used open field and tonic immobility tests. The open field test showed indoor hens were more fearful, the tonic immobility test did not differentiate the ranging groups, and the attention bias test had limited application to more behaviourally extreme test subjects as it relied on hen movement. Alternatively, latency to eat food in a novel testing environment (with no alarm call playback) including habituation across time was informative in differentiating test groups. If hens in a novel arena are diverting their attention towards their surroundings versus eating then this could be used as a measure of attention bias that is less selective on individual hen inclusion although differences in food motivation may need to be considered.

Indoor hens were also more anxious than outdoor hens as indicated by a reduced number of individuals who were willing to eat following the playback of the alarm call suggesting the hens were showing an attention bias towards their surrounding environment rather than

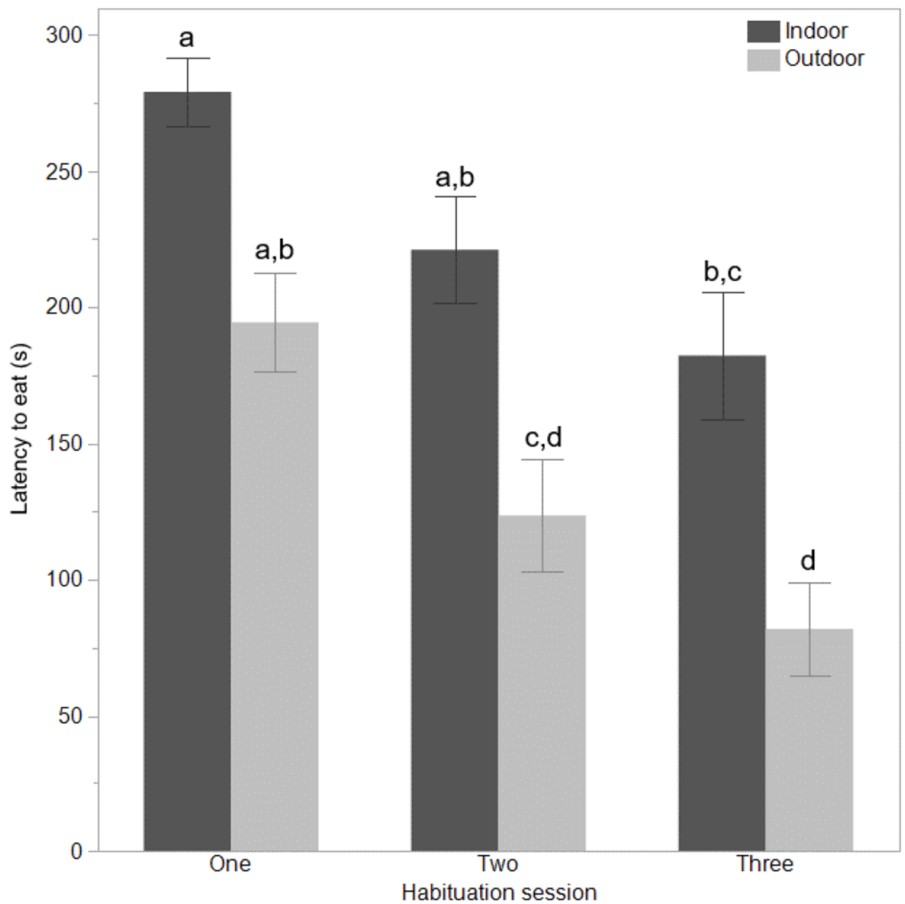

**Figure 2** **Time to eat for indoor and outdoor free-range hens.** The mean ± SEM latencies (seconds) to first eat across three habituation sessions in a test arena for hens that ranged daily (outdoor), or did not range (indoor). Dissimilar superscript letters (a-d) indicate significant differences between ranging groups across habituation sessions ($P < 0.008$).

**Table 2** **The numbers of indoor and outdoor hens that ate during an attention bias test.** The number and corresponding percentages of the hens from different range use groups (indoor, outdoor) that did not eat at all, and did not or did eat after the alarm call playback in an attention bias test.

|  | Did not eat No alarm call | Did not eat after alarm | Ate after alarm |
|---|---|---|---|
| Indoor | 13/28 = 46% | 14/15 = 93% | 1/15 = 7% |
| Outdoor | 6/34 = 18% | 17/28 = 61% | 11/28 = 39% |

the positive stimulus of food. However, fewer indoor hens reached the criterion of eating at all to receive an alarm call playback, and thus the sample size for the attention bias test was greatly reduced. The typical motionless response of laying hens in novel environments highlights a confound with individual behavioural assessments that requires movement to execute the test. Previous measurements of cognitive and learning abilities that require bird training have excluded hens that do not make training criterion due to lack of movement (*Campbell et al., 2018*; *Krause et al., 2006*; *Nordquist et al., 2011*). Similarly, previous tests

of judgement bias have shown fearful hens were less flexible during task learning than non-fearful hens (fearfulness was assessed in an open field test; *De Haas, Lee & Rodenburg, 2017*). Unfortunately, these non-moving/learning individuals are likely informative to the questions being asked, but logistically cannot be assessed further, removing potentially vital subjects. It was anticipated that the older age of the hens in the current study may have resulted in rapid acclimation to the testing arena as seen in free-range hens of a similar age tested in *Campbell et al. (2019)*. This would enable a once-off attention bias test making it a potentially rapid test of anxiety for measuring laying hen welfare, similar to the test applied to sheep and cattle (*Lee et al., 2016*; *Lee et al., 2018*; *Monk et al., 2018a*). But the lack of movement during the open field test indicated the birds would require more time in the arena to become comfortable enough to eat. Each hen was habituated to the arena in the presence of food, extending the time and labour required to execute the attention bias test. However, even after a total of 20 min in the testing arena across four separate sessions, many hens (predominately indoor hens) were still unwilling to eat. Comparatively, the habituation sessions in *Campbell et al. (2019)* totalled 30 min in the arena and all birds were rapidly eating following placement, but these were all indoor-housed hens and not from specific treatment groups prior to the drug administration. The indoor hens in the current study were likely representative of extreme fearfulness/anxiety within the study population and it is unclear if they would have habituated over further sessions in the arena or if they would have remained unwilling to eat. Male Swedish bantams tested in a 'startle' test (*Favati, Leimar & Lovlie, 2014*) were first observed in a novel arena for 20 min before playback of the alarm call, giving them time to adapt to the environment. The lack of adaptation by the indoor hens in the current study was, however, informative in itself. We suggest the willingness to eat in a novel, isolated environment is an alternative measure of attention bias where the unfamiliar environmental surroundings are the perceived threat and this test could be applied once, or across time to categorise anxiety in hens. Alternatively, each bird could only be tested once, but for an extended period. Consecutive time in the test arena rather than multiple habituation sessions may increase the rate of adaptation, but this remains to be confirmed.

Potential differences in food motivation could have also played a role in these trials if the outdoor hens were hungrier from ranging versus staying indoors where the food was located. The majority of hens were captured from inside prior to testing, suggesting that indoor and outdoor hens would have had a similar opportunity to eat prior to testing. Previous testing in a T-maze with a food reward showed indoor hens were slower to learn to access the food reward and both indoor and outdoor hens were kept inside during testing days and food restricted (*Campbell et al., 2018*). However, it is unknown if there were inherent differences in appetite between the two ranging groups based on their established daily behavioural patterns (*Kolakshyapati et al., 2019*). If outdoor hens were chronically feed 'restricted' because of their choice to range outdoors this may have affected general activity levels in the test arena and/or motivation to consume the grain. Further development of an attention bias test using a food reward could first assess for any differences in food motivation between treatment groups.

To minimise the impacts of novel testing environments, an alarm call could be played to hens in their home cages. *Brilot & Bateson (2012)* played an alarm call threat to starlings in their home cages to demonstrate a reduced willingness to feed and increased vigilance in birds without water bath access, indicating the importance of bathing to the starling's welfare. A similar method could thus be applied to caged laying hens or small groups of hens to observe individual-level responses (*Brendler, Kipper & Schrader, 2014*) but larger-groups of hens would likely prevent accurate individual-level observations. A visual predator stimulus could also be used (*Schütz, Forkman & Jensen, 2001*) for which presentation to individual birds could be better controlled than auditory playbacks but could also be applied at a group level (*Zeltner & Hirt, 2008*). Caution would have to be placed on ensuring the stimulus was threatening to the hens. An inanimate model owl was presented to some of the hens in the current study to assess if this may be a better threat than an alarm call playback but no visible reaction from the hens could be detected (these hens were excluded from the subsequent auditory playback testing). Ultimately individual hen responses may depend on where they are tested (*Graml, Niebuhr & Waiblinger, 2008*) and the decision for individual testing in isolation or not could be factored into the research question of interest. Similarly, a different positive stimulus such as a foraging/dust bathing substrate could be used to avoid any confounds of appetite on bird responses to food. Attention bias testing in sheep replaced food with an image of a conspecific to mitigate the impact of appetite on behavioural responses (*Monk et al., 2018b*). But caution would still need to be placed on the influence of prior activity in the home pen for litter-housed birds and individual differences in motivation to utilise resources. Deprivation of resources other than food can lead to similar differences in exploratory behaviour compared with non-deprived individuals (*Nicol & Guilford, 1991*). Finally, the test applied in this study is specific to the context of one threatening situation and other forms of attention bias tests with different types of cues may yield differing results between treatment groups (*Crump, Arnott & Bethell, 2018*).

In the current attention bias test, measures of latencies to perform locomotory, vocal, or appetitive behaviours were included. In *Campbell et al. (2019)*, the older tested hens displayed clear vigilance behaviour where they stretched their head and neck, this same type of vigilance was not displayed in the younger birds and was not observed in the current study. Other tests have included measures of vigilance but of slightly varying definitions (e.g., *Beauchamp, 2019*; *Brendler, Kipper & Schrader, 2014*; *Favati, Leimar & Lovlie, 2014*; *Odén et al., 2005*; *Zidar & Løvlie, 2012*) which may make it difficult to compare between studies if this is included as a measure in an attention bias test. If vigilance is defined as head upright and hen alert (rather than neck stretching), then this is likely a directly opposing behaviour to the time spent eating in the arena (*Beauchamp, 2019*; *Favati, Leimar & Lovlie, 2014*). Time spent eating was included in the pharmacological validation test in *Campbell et al. (2019)* but not in the current study as fewer birds ate. Latencies are standard measures to include that are easily comparable across testing situations with different hen groups.

Indoor hens were demonstrated to be more fearful than outdoor hens. This corresponds with previous tests of fear across several sample groups of free-range hens (*Campbell et al., 2016*; *Hartcher et al., 2016*; *Larsen et al., 2018*). Differing range use groups have

previously been assessed using open field, tonic immobility, novel object, and human approach/avoidance tests (*Campbell et al., 2016*; *Hartcher et al., 2016*; *Larsen et al., 2018*; *Mahboub, Muller & Von Burrell, 2004*). The open field tests demonstrated that indoor hens were more likely to reduce their movement and were slower to vocalise suggesting they were fearful of the new environment and were motionless and quiet to avoid predator detection (*Gallup & Suarez, 1980*). However, similar to previous studies with free-range hens (*Campbell et al., 2016*; *Larsen et al., 2018*) and other poultry treatment groups (e.g., hens: *Fraise & Cockrem, 2006*), quail: *Miller, Garner & Mench, 2006*), not every applied test differentiated the hens. The duration of tonic immobility did not differentiate indoor and outdoor hens in this current study, nor the ranging groups of *Campbell et al. (2016)* or *Mahboub, Muller & Von Burrell (2004)* and only differentiated hens within one flock of *Larsen et al. (2018)*. In contrast, *Hartcher et al. (2016)* used only the tonic immobility test to show hens with longer tonic immobility spent less time outdoors. Differences between studies could be attributed to a multitude of variation including the precise method of conducting the tonic immobility test, hen age, strain, and life histories (*Jones, 1986*). However, the results from this and previous studies suggest that assessments of fear in laying hens should include multiple different tests. Tonic immobility tests an anti-predator response and thus may be less sensitive to environmental differences suggesting this test alone may be insufficient to determine variation in fear or not.

The indoor hens in the current study also showed feather damage. The extent of damage was confounded with range use for the selected hen groups making it difficult to determine the role of feather damage in the levels of fear, and the causative relationship with range use. Previous studies have found better plumage condition in hens scored while outdoors compared to hens scored while indoors (*De Koning et al., 2018*), better plumage in those hens that ranged farthest (*Chielo, Pike & Cooper, 2016*) or better plumage in hens tracked to range more frequently (*Rodriguez-Aurrekoetxea & Estevez, 2016*). But not all studies have found associations between ranging and plumage condition (*Hartcher et al., 2016*; *Larsen et al., 2018*). In some previous behavioural tests with free-range hens, all tested hens were of good plumage condition, and in overall visibly good health (*Campbell et al., 2016*) but this could have been related to the younger age (37-39 weeks of age) and a relationship with plumage may have still developed later in the production cycle. There is extensive research demonstrating an association between fear and feather pecking in hens but the exact nature and direction of the relationship is complex (e.g., *Uitdehaag et al., 2008*; *Van der Eijk et al., 2018*. The causative nature of the relationship between fear, range usage, and plumage condition was unable to be determined from the current study but warrants further investigation.

## CONCLUSIONS

Attention bias tests could be used to assess the negative affective state of anxiety in laying hens, but the application of a test that requires the hens to move is limited for extremely anxious/fearful individuals. Measuring willingness to feed in a novel environment is informative and could be applied to all individuals; however, assessment of differences in

food motivation between treatment groups should also be included. Multiple behavioural tests are recommended when assessing hen states as not every test will differentiate hen treatment groups.

## ACKNOWLEDGEMENTS

We are grateful for the technical and data collection assistance of Andrew Cohen-Barnhouse (University of New England), Md Saiful Bari (University of New England), and Tim Dyall (CSIRO).

### Funding

Poultry Hub Australia providing funding for the research (grant number 2017-20). Emily Dickson was supported by a Commonwealth Scientific and Industrial Research Organisation (CSIRO) summer vacation student scholarship. The funders had no role in study design, data collection and analysis, decision to publish, or preparation of the manuscript.

### Grant Disclosures

The following grant information was disclosed by the authors:
Poultry Hub Australia: 2017-20.
Commonwealth Scientific and Industrial Research Organisation (CSIRO).

### Competing Interests

The authors declare there are no competing interests.

### Author Contributions

- Dana L.M. Campbell conceived and designed the experiments, performed the experiments, analyzed the data, contributed reagents/materials/analysis tools, prepared figures and/or tables, authored or reviewed drafts of the paper, approved the final draft.
- Emily J. Dickson performed the experiments, analyzed the data, authored or reviewed drafts of the paper, approved the final draft.
- Caroline Lee conceived and designed the experiments, contributed reagents/materials/-analysis tools, authored or reviewed drafts of the paper, approved the final draft.

### Animal Ethics

The following information was supplied relating to ethical approvals (i.e., approving body and any reference numbers):

The University of New England Animal Ethics Committee approved the research (AEC17-092).

### Data Availability

Data are available in the CSIRO Data Access Portal: Campbell, Dana; Dickson, Emily; Dyall, Tim; Lee, Caroline (2019): Behavioural test data. FreeRange layers. v2. CSIRO. Data Collection. https://doi.org/10.25919/5d75e59e8908c

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
