# Peer review of "Application of open field, tonic immobility, and attention bias tests to hens with different ranging patterns"

_PeerJ, doi:10.7717/peerj.8122_

## Round 0.1 · original submission · Minor Revisions

This is an interesting piece of work. The manuscript will be improved by responding to the questions about methodology raised by Reviewer 1, and altering Figure 1 as suggested.

Reviewer 2 asks you to consider whether differences in feed motivation could partially underlie the results obtained from indoor and ranging birds. Might ranging birds also have had greater prior experience of alarm calls? Are alarm calls given more frequently on the range than indoors? Could this result in any habituation or sensitisation?

It would also strengthen the paper to ensure that the introduction does not give an overly optimistic (!) view of cognitive/attention bias tests conducted so far, as results with farm animals have been so mixed and can be difficult to interpret. Contradictory or "no effect" outcomes could be added as a concern on line 92. How do the current results fit with your previous findings of no bias in socially isolated hens (Hernandez et al..2015). Even work using pharmacological methods can produce confounds. Destrez et al (2012) found that anxiolytic drugs in sheep may have reduced anxiety but also interfered with learning in a CB test. Does the method you use avoid such difficulties? Have attention bias studies produced fewer contradictory or "no effect" outcomes than CB studies?

It would be good to cite other chicken studies on CB or attention bias where fearfulness or anxiety has been the emotional state of interest. Please consider your findings alongside those of e.g. Salmeto et al. (2011) Brain Research (I think they used a predatory stimulus) and also de Haas et al. (2017) Frontiers in Vet Sci, who found that more fearful birds had a more optimistic bias. It is important for the advancement of this field that reasons for different findings are explored across studies.

·

Basic reporting

New technology has become available to monitor animals in large groups on an individual level and we are puzzled by the extent of individuality e.g. in “simple” animals like chickens. In most cases we have no idea how this variation in e.g. ranging patterns comes about and which trait(s) underlie this variation i.e. cause it. This study is not the first to apply behavior tests to laying hens with different ranging patterns but the valuable thing about the attention bias test is that it was pharmacologically validated. It also has some biological meaning for laying hens because they (including their wild ancestors) always face the dilemma between being safe and starve or taking risks and find food. Therefore, this is a very interesting, valuable study adding new knowledge of the (ranging) behavior of laying hens and possible underlying personality differences.
Fear tests often face the problem of ceiling effects: The stimulus is too strong or too weak so the results do not distinguish the tested animals satisfactorily. This is also the case in this study to some extent because the indoor birds were too reluctant to feed in most cases. The limitations of this study are discussed by the authors.
For the most part the manuscript is understandable and conforms to the standard of a high-ranking scientific journal. Ideas for improving the figures and clarifying some parts of the text are listed below. Raw data are supplied in a clear format.

Experimental design

The statistical analyses seem sound and appropriate for the dataset.
However, the selection of the hens remain unclear: Was the selection of hens roughly equal between the 3 pens or was mainly one pen chosen? I am asking that because often pens differ how many hens range. So, it could be that in the high ranging hen group you only picked birds from one pen and in the low ranging hen group you exclusively or mainly picked hens from another pen. This would be important for the interpretation of the results

Line 140: Why did you select 67 hens? How did you choose this sample size?

Line 198: Adding to the point about choosing your focal hens your selection based on plumage could influence the results. There are reports in the literature that birds with plumage or skin damage are more stressed (e.g. Krause et al., 2011). If you could not make subgroups due to plumage quality, why did you not choose your birds randomly? You confounded ranging behavior with plumage quality (later mentioned in the results).

Why did you choose unequal numbers of indoor and ranging hens?

Line 249f: Why did you use 2 different calls?
Lines 270ff: It is logical that there was an effect (or trend) for the indoor birds and not the outdoor birds because the variation in plumage quality was larger in the outdoor birds. Is this the reason why you tested the 2 groups in 2 separate analyses?
Line 276: But you confounded feather score with ranging in that way. You should have used feather score as a covariate instead.
Line 300: Were these 48 birds all birds that ate or how was the sample size reduced to 48?

Line 335: Why was the total number of indoor birds lower than that of outdoor birds?
Did you test whether the number of birds feeding in the novel arena or the number of birds feeding after the alarm call differed among the groups?

Validity of the findings

The results are well discussed in a meaningful way where inconclusive results are acknowledged.

Raw data are supplied in a clear format.

Additional comments

Lines 59f: This is not a good summary of the results. Latency could not be measured in many birds so you should rather write that than that this was a poor marker.
Line 118: Zeltner and Hirt also used a hawk model: Zeltner, E.; Hirt, H. (2008): A note on fear reaction of three different genetic strains of laying hens to a simulated hawk attack in the hen run of a free-range system. Applied Animal Behaviour Science 113, pp. 69–73.
Animals and Housing
There is a discrepancy about housing conditions. First the authors write that the birds were housed in the same conditions as during rearing (Line 147) and that they only used birds from the control = just litter no perches. But then the pens are described to contain perches (line 152). Was litter also provided?
RFID: What was the frequency?
Line 145: It is not clear whether in the pens with novel objects there were the perches, was there one room with novel objects, one room with perches, one room with controls or were they mixed?
Line 151: Did the pens have litter?
Line 159: Did the outdoor range have vegetation?
Lines 204f: How could the birds range during testing times? When they were tested, they were confined to the test box.
Test arena: Was there light to facilitate the videos in the testing arena? What was the light level in the testing arena? Was it comparable to the light level in the barn? Barns are much darker than the range and ranging birds might be more used to bright lights than birds that always remain in the barn.
Line 221: The open field test was repeated or why did you count the number of steps during habituation?
Line 227: Were the pen feeders in the barn? Did you watch whether all test birds ate from those feeders? Otherwise not all hens were acclimated to the novel food which could be a problem.
Line 236: To clarify, I would add ‘before habituation’ after ‘achieving this’.
Alarm calls: How did you make sure that the non-tested birds did not hear the alarm calls?
Line 249f: Why did you use 2 different calls?
Line 300: Were these 48 birds all birds that ate or how was the sample size reduced to 48?
Fig. 1 It makes no sense to connect the points because session is a distinct category. Instead of using the lines it would be better to have the categories indoor/outdoor on the same line to compare the latencies between the 2 groups. This is not so easy now. The arrangement in Fig. 2 is much clearer.
Line 335: Why was the total number of indoor birds lower than that of outdoor birds?
Did you test whether the number of birds feeding in the novel arena or the number of birds feeding after the alarm call differed among the groups?
First paragraph of discussion: I agree that the fact whether a bird ate food in a novel environment is a useful parameter to distinguish the groups but why do you call this a measure of attention bias?
Line 354: Could you control hunger state? Indoor birds are always close to the feeding trough whereas ranging birds might be hungrier. It should be mentioned that the attention bias test implies that the hunger state was constant among birds.
Line 467: Albentosa et al. is not cited in the text.

·

Basic reporting

The study is clearly described and presented and well-placed within the context of the previous literature.

It would be useful to know which behavioural tests were used in the previous literature to classify indoor preferring hens as more fearful, since your validation might also include these tests.

I have no other comments about this section.

Experimental design

The study is generally rigorously performed and described. There is a pressing need to identify methods to assess subjective state in animals and the study is a welcome addition to the scientific literature in this area.

The main issue that concerned me is the use of measures relating to feeding to make inferences about animals' subjective state. The authors state that latency to eat in a novel arena is an informative measure of anxious state. I would propose that the two groups used in the study: indoor and outdoor-preferring hens, are likely to also differ in their motivation to access food. It is suggested in the methods that the ad libitum feed is provided in the litter pens, and not in the range. It therefore seems likely that 'indoor' birds would have fed more recently than 'outdoor' birds due to their proximity to the feeder. This would reduce indoor birds' motivation to access food in the test. Secondly, I would hypothesise that 'outdoor' birds might be more motivated to access food due to their more active lifestyle. These two factors might cause differing food motivation in the two groups, compounding their responses to the behavioural test. I would like to know more about the food motivations of the two groups e.g. were any food motivation tests performed or was access to food monitored or controlled during the pre-test period? Can the authors discuss the influence of food motivation on attention bias and other behavioural tests.

Validity of the findings

The authors provide a great discussion of the limitations of behavioural tests that incorporate the requirements of movement and in the use of subjects at both ends of a behavioural spectrum (high and low fearfulness). The study is a great example of the limitations of such an approach and the authors provide a comprehensive discussion of this.

However, I have concerns about the main finding that latency to eat in a novel arena is an informative measure of emotional state due to the unknown underlying food motivations in the two groups of birds (as explained in the previous section). I would ask that the authors provide more information about whether/how food motivation was assessed in the pre-test period and/or discussion of the limitations of a feed-based approach.

Additional comments

At the beginning of the discussion, could you remind the reader of the aims of the study.

I wonder if the tonic immobility test could not differentiate between the groups in this study and previous studies, because it is a measure of an antipredator response which may be relatively inelastic to behavioural differences between the two groups? In contrast the novel arena test measures response to a less severe threat (and one that we already know the indoor preferring birds are likely to respond negatively to).

---

## Round 0.2 · accepted · Accept

A considered response to all reviewers' comments, with much clearer explanation of potentially confounding factors.